# An Academic and Technical Overview on Plant Micropropagation Challenges

**Neama Abdalla** [1,2,*] , **Hassan El-Ramady** [3,*] , **Mayada K. Seliem** [4] , **Mohammed E. El-Mahrouk** [5] , **Naglaa Taha** [6] , **Yousry Bayoumi** [5] , **Tarek A. Shalaby** [5] and **Judit Dobránszki** [2]

1  Plant Biotechnology Department, Biotechnology Research Institute, National Research Centre, 33 El Buhouth St., Dokki, Giza 12622, Egypt
2  Centre for Agricultural Genomics and Biotechnology, FAFSEM, University of Debrecen, 4400 Nyíregyháza, Hungary; dobranszki@freemail.hu
3  Soil and Water Department, Faculty of Agriculture, Kafrelsheikh University, Kafr El-Sheikh 33516, Egypt
4  Ornamental and Floriculture Department, Horticulture Research Institute, Antoniadis, Alexandria 21599, Egypt; mayadaseliem@gmail.com
5  Physiology & Breeding of Horticultural Crops Laboratory, Horticulture Department, Faculty of Agriculture, Kafrelsheikh University, Kafr El-Sheikh 33516, Egypt; threemelmahrouk@yahoo.com (M.E.E.-M.); ybayoumi2002@yahoo.com.sg (Y.B.); tashalaby@yahoo.com (T.A.S.)
6  Plant Pathology Research Institute, Agriculture Research Center, Giza 12619, Egypt; naglaa_abdelbaset@yahoo.com
*  Correspondence: neama_ncr@yahoo.com (N.A.); hassan.elramady@agr.kfs.edu.eg (H.E.-R.)

**Abstract:** The production of micropropagated plants in plant-tissue-culture laboratories and nurseries is the most important method for propagation of many economic plants. Micropropagation based on tissue-culture technology involves large-scale propagation, as it allows multiplication of a huge number of true-to-type propagules in a very short time and in a very limited space, as well as all year round, regardless of the climate. However, applying plant-tissue-culture techniques for the commercial propagation of plants may face a lot of obstacles or troubles that could result from technical, biological, physiological, and/or genetical reasons, or due to overproduction or the lack of facilities and professional technicians, as shown in the current study. Moreover, several disorders and abnormalities are discussed in the present review. This study aims to show the most serious problems and obstacles of plant micropropagation, and their solutions from both scientific and technical sides. This review, as a first report, includes different challenges in plant micropropagation (i.e., contamination, delay of subculture, burned plantlets, browning, *in vitro* rooting difficulty, somaclonal variations, hyperhydricity, shoot tip necrosis, albino plantlets, recalcitrance, shoot abnormalities, *in vitro* habituation) in one paper. Most of these problems are related to scientific and/or technical reasons, and they could be avoided by following the micropropagation protocol suitable for each plant species. The others are dominant in plant-tissue-culture laboratories, in which facilities are often incomplete, or due to poor infrastructure and scarce funds.

**Keywords:** acclimatization; browning; contamination; delay of subculture; hyperhydricity; recalcitrance; somaclonal variations; totipotency

## 1. Introduction

Plant-tissue culture (PTC) refers to the *in vitro* culture of plant cells, tissues, organs, seeds, protoplasts, or embryos on a nutrient medium under aseptic conditions, where temperature, photoperiod, humidity, light, and the components of medium all supply the ideal, controlled growing environment [1]. PTC, as a crucial component of plant biotechnology, was and still is one of the optimum approaches, which can be applied to overcome many problems (i.e., global warming, climate change, desertification, salinization, the global water crisis) faced by agricultural and horticultural production, which are causing a global food crisis and famine [2,3]. According to the United Nations, the population of the

world will increase to 9.9 billion by 2050, thus, food production must rise by 50% to feed billions more people. PTC can guarantee continuous production systems regardless of the environmental or geographical constraints to meet the food, feed, fiber, and energy supply needs of a growing population [2]. Several promising PTC techniques have been used, including *in vitro* micropropagation, organogenesis, and somatic embryogenesis [4]. PTC also has many benefits, such as production of pathogen-free plants [5,6]; somatic hybridization [7,8]; rapid propagation of difficult-to-propagate plants [9]; improving genetics of commercial plants as desired [10]; obtaining androgenic and gynogenic haploid plants for shortening breeding programs [11]; conserving rare and endangered plants [12]; producing different varieties tolerant to abiotic stresses such as drought, salinity, and heat [13,14]; and producing biological active compounds or secondary metabolites, especially through plant-cell-suspension culture [15,16].

PTC is considered the backbone of horticultural nurseries that can be used in propagating plants of forestry, vegetable, fruit, and ornamental species [5]. These nurseries have huge economic importance for many horticultural production systems, e.g., seedling production for tree-growing programs [17]; availability of high-quality tree planting materials [18]; and disease screening and discovery of new plant pathogens to maintain healthy nurseries [19], with the aim of producing high-quality, productivity as well as pathogen-free plants and/or trees. However, there are many obstacles, which may cause a loss for *in vitro* cultured plants in nurseries and *in vitro* culture laboratories, such as contamination of cultures [20], hyperhydricity phenomenon [21], browning of tissues or phenols exudation [22], shoot tip necrosis [23], delay of subculture, somaclonal variations [24,25], root hardening [20], and failure of acclimatization or limited planting [26].

In addition, micropropagation technology is more expensive than the traditional methods of plant propagation, where it is a capital, labor, and energy-intensive industry, and the unit cost per plant becomes unaffordable as well in some cases. Moreover, it requires many types of technical skills. In many developing countries, the financial resources of equipment, tissue-culture facilities, and trained personnel are often not easily available. Moreover, there is the energy needed for tissue-culture technology, particularly electricity, which is important for controlling environmental conditions such as temperature, day length, and relative humidity, and the clean water, which are costly [27]. Therefore, this review focuses on the different reasons that may cause losses in the production of tissue-cultured plants in laboratories. Then, it presents suggestions and solutions to overcome these problems.

## 2. Methodology of the Review

A large and growing body of the literature has addressed the various problems of micropropagation and its challenges in PTC laboratories. This was the starting point of the current review. The most important successful points in any literature review mainly depend on the selected sections in the article and the harmony among these sections. So, a proper table of contents (TOC) was established, and precise keywords were selected based on the title of the MS. The main sections in the recent MS were the challenges and problems of plant micropropagation, from one side, and the disorders and abnormalities of it, from the other side. Therefore, the selected keywords for searching in this MS were "Contamination", "Delay of subculture", "Burned plantlets", "Browning", "*In vitro* rooting difficulty", "Failure of subsequent acclimatization", "Hyperhydricity", "Shoot tip necrosis", "Albino plantlets", "Recalcitrance in clonal micropropagation", "Shoot abnormalities", and "*In vitro* habituation". After building the suggested TOC and preparing the suggested keywords, the main publishing websites were visited to collect the desirable and sound published materials, including articles, reviews, mini-reviews, chapters and books. The most visited websites were ScienceDirect, MDPI, Frontiers, SpringerLink, PubMed, etc. What about the criteria of selecting articles for this MS from previous websites? The reputation of both the selected journal and authors was the main reason for selection. Due to plant-tissue culture having a great commercial dimension, so many secrets in published

articles in this field are not announced, our co-author has fortunately served in many companies for plant biotechnology. So, this MS included information from published articles besides some expertise from the side of working in the field of PTC. Therefore, this is the first report that involved all the studied plant micropropagation problems, which are common in most laboratories, in one article, and their suggested solutions depending on both the academic (scientific) concepts and technical (applied) expertise.

## 3. Problems Associated with Plant Micropropagation

Any PTC laboratory needs some basic facilities or conditions, without them it is impossible to produce any micropropagated materials. These include the complete infrastructure and trained workers for proper controlled-environmental conditions [28]. The success of any PTC laboratory producing large-scale plant material depends mainly on these previous factors, besides the scientific team and their efficiencies (Figure 1). Making great progress in the different branches of biotechnology, there is a terrible race between different mega-companies and scientific centers in developing laboratories for PTC [28]. This reflects the significance of these laboratories and their role in the global bioeconomy.

Due to the great potential of PTC in both scientific research laboratories and commercial companies, several applications for this vital field could be achieved, but this sector still face a lot of problems, especially regarding plant micropropagation, which causes a lot of economic crises for these laboratories and companies. These problems are highlighted in the following sections in this review, with coverage for all those difficulties and their scientific explanations, starting with laboratory construction, as well as solutions that shed light on the elucidation of all the complications of this technique [29].

### 3.1. Problems Originated Due to Technical Reasons

3.1.1. Contamination of Plant-Tissue Cultures

The contamination of *in vitro* plants is considered a crucial obstacle, which prohibits successful micropropagation protocol (Figure 2). Contamination may include many microorganisms, such as bacteria, fungi, molds, and yeasts. This contamination is the main factor in the losing of time and effort related to PTC, which increases the cost of production [27]. External contamination results from the laboratories and used materials (media; glassware; culture vessels, tools, explants), whereas internal contamination is related to the endophytic microbes in mother plants [30]. Several proper methods could be used to exclude and eliminate the contaminants through surface sterilization (Table 1), such as chemical agents (antiseptic agents, liquid detergent, mercuric chloride or sodium hypochlorite), ultraviolet (UV) sterilization, autoclaving of media and instruments, and improvement of cultural practices or handling [20]. Therefore, surface sterilization of the equipment and plant materials should be managed to improve the performance of the laboratories and, thus, acquire aseptic cultures [30]. On the other hand, antibiotics could be used as anti-microbial agents for eliminating endophytic bacteria in *in vitro* cultured plants [31].

In this regard, banana axillary shoots contaminated by internal bacteria were cultured on MS medium supplemented with different filter-sterilized antibiotics (ampicillin, penicillin, ticarcillin) added separately at various concentrations (25, 50, 100, 200 mg $L^{-1}$). The results showed that the studied antibiotics recorded zero contamination at 100 or 200 mg $L^{-1}$, however, a reduction in shoot-multiplication parameters was noticed. The most effective concentration of a single added antibiotic for eliminating the bacterial contamination was 100 mg $L^{-1}$ [43].

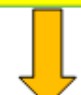

**Figure 1.** The basic information about the field of plant-tissue culture, including the definition, different applications, advantages, and disadvantages of this field. Sources: [9,32].

**Table 1.** Some surface-sterilization methods for external contamination used in tissue-culture laboratories.

| Studied Plants | Surface-Sterilization Methods and Sterilized Items | Main Findings of the Study | Refs. |
|---|---|---|---|
| Potato (*Solanum tuberosum* L.) | Ultraviolet-C radiation to explants for 10 min | The external contamination of explants has been reduced. | [29] |
| Mulberry (*Morus alba* L.) | Mercuric chloride at 0.2% for 10 min to shoot tips and auxiliary buds | Minimum percentage of contamination and highly percentage of survival in correlation with shoot development were noticed for all mulberry cultivars under investigation. | [33] |
| Guayusa (*Ilex guayusa* Loes.) | 70% ethanol for 2 min + 2.5% sodium hypochlorite + five drops of Tween-20 for 25 min to apical stem segment | Surface sterilization with ethanol and sodium hypochlorite resulted in 100% surface-sterilized stakes. | [34] |
| Eucalyptus (*Eucalyptus obliqua*) | Active chlorine added to culture media at 0.005% for establishment and 0.003% for shoot multiplication and elongation | Active chlorine at 0.005% led to the lowest fungal contamination rate in the establishment stage, where at 0.003% resulted in maximum number of shoots per explant and the greatest shoot length in the multiplication stage. | [35] |
| Guava (*Psidium guajava* L.) | Silver nanoparticles (AgNPs) at 50 mg $L^{-1}$ directly to shoot tips for 5 min or at 5 mg $L^{-1}$ added to culture medium | AgNPs at 50 mg $L^{-1}$ yielded a contamination rate of 40%, where at 5 mg $L^{-1}$ reduced shoot contamination rate to 50% compared to controls (80%) and enhanced multiplication rate of the shoots by 180%, as an alternative method for surface sterilization of explants which are easily damaged by commonly used surface sterilizing. | [36] |
| *Rosmarinus officinalis* L. | Bio-synthesized of silver nanoparticles via *Rubia tinctorum* L. using cell culture were applied for surface sterilization of stem explants | Sterile explant percentages varied between 40 and 97% and no browning was observed. This method could be used in surface sterilization of explants which have a browning problem caused by their phenolic contents. | [37] |
| *Sargassum fusiforme* | A crude extract of a medicinal herbal plant Tarragon (*Artemisia dracunculus*) was used for surface sterilization of explants (leaf, stipe, and stolon) cultured *in vitro* | The crude extract of A. *dracunculus* showed a high microbial sterilization effect with (90, 80, and 20%) for leaves, stipes and stolons, respectively. It has very low toxicity to plant tissues compared to chemical sterilants. | [38] |
| Chinaberry (*Melia azedarach* L.) | Dipping leaf explants in 2 g $L^{-1}$ benomyl for 2 h + 7% hydrogen peroxide ($H_2O_2$) for 10 min + 2% NaOCl for 12 min for surface sterilization | The lowest contamination percentage of explants and browning as well as the highest percentage of callus induction and growth were observed. | [39] |
| Carnation (*Dianthus caryophyllus*) | Sodium dichloroisocyanurate (NaDCC) was applied as a medium sterilizer to culture medium at 0.02 g $L^{-1}$ instead of autoclave sterilization | Contamination rate recorded below 5%, sodium isocyanurate has the potential to substitute media autoclaving in plant-tissue culture. | [40] |
| Butterfly pea *Clitoria ternatea* L. | 0.1% Bavistin solution + 70% ethanol, and 0.1% $HgCl_2$ was used for surface sterilization of nodal explants | Microbial contamination was eliminated and then surface sterilized nodal explants were used for shoot multiplication induction. | [41] |
| Orchid (*Angraecum rutenbergianm* Kraenzl) | 0.5% (*w/v*) NaDCC solution + 2 mL $L^{-1}$ Plant Preservative Mixture (PPM™) was used for surface sterilization of Seed capsules | 87.5% of the total number of capsules was disinfected and the seeds inside them were clean after 3 months of culture. Both NaDCC and PPM were essential to suppress microbial growth. | [42] |

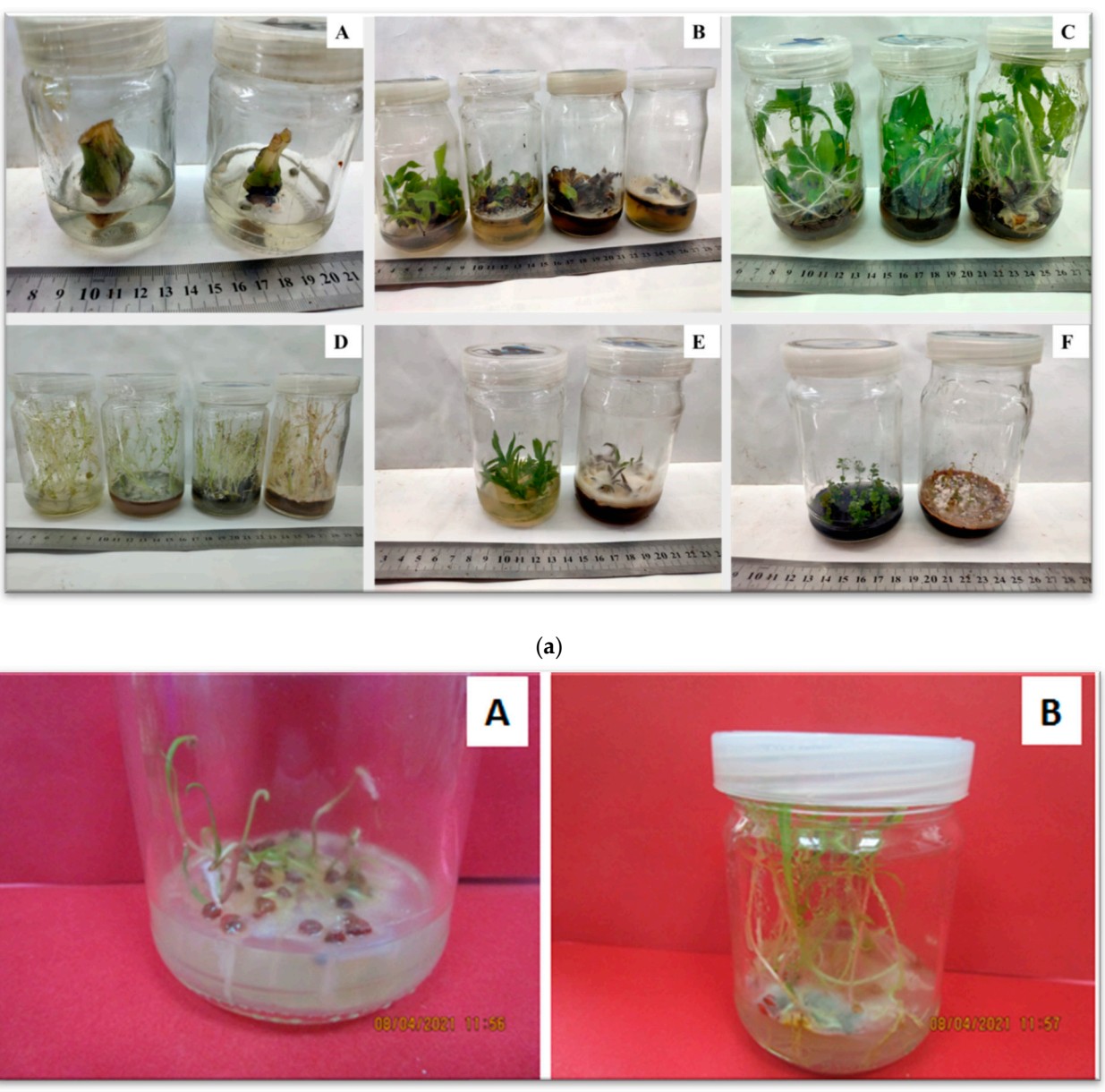

**Figure 2.** (**a**) Effect of contamination on *in vitro* plants (left jar = normal plant, right jar or jars = contaminated plant). (**A**) Banana plant (*Musa* sp.) at initiation stage of meristem culture, (**B**) banana plant at shoot multiplication stage, (**C**) banana plant at rooting stage, (**D**) potato plant (*Solanum tuberosum* L.), (**E**) *Cattleya* sp., and (**F**) blueberry (*Vaccinium corymbosum* L.) at rooting stage. (**b**) Effect of contamination on *in vitro* growth and shoot proliferation of spinach plant (*Spinacia oleracea* L.) (**A**) = Bacterial contamination and (**B**) = fungal contamination.

The external contamination by epiphytic microbes could be inhibited by surface sterilization with running water with/without detergent, chemical substances (ethanol, mercuric chloride, sodium hypochlorite), and plant-preservative mixtures [29]. However, there are some materials could be added to the culture media to inhibit the external contamination, such as plant-preservative mixtures [29] and benomyl fungicide [39]. The investigation showed that inclusion benomyl at 100 and 500 mg L$^{-1}$ in culture medium significantly decreased the fungal-contamination percentage in *Melia azedarach* L. *in vitro* cultures from leaf explants, minimized browning, and recorded the highest percentage of callus induction and growth [39]. Endophytic microbes that are present within the

explants are considered a major constrain to the establishment and growth of tissue-cultured plants, as they are more difficult to remove by normal surface sterilization. However, the internal bacterial contamination could be eliminated by supplying the culture media with different substances (Table 2), for example, antibiotics [44,45], copper sulfate [44,46], or fungicides [47]. After identification of the contaminants, a low-phytotoxicity antibiotic should be selected [47].

**Table 2.** Some antiseptic substances used for eliminating endophytic contamination in plant-tissue cultures.

| Applied Substance (Name and Concentration) | Plant Species/Cultivar | Success of Decontamination | Refs. |
|---|---|---|---|
| Kanamycin and streptomycin sulphate at 10 $\mu$g mL$^{-1}$ each were added to shoot multiplication medium +2 mg L$^{-1}$ BAP +10 mg L$^{-1}$ adenine sulfate | *Guadua angustifolia* Kunth | Bacterial growth was inhibited and intensive formation of high-quality shoots was observed. | [48] |
| Antibiotics (timentin at 150 mg L$^{-1}$ + gentamycin at 30 mg L$^{-1}$ were added to culture medium | *Camellia sinensis* var. *sinensis* | They were effective to eliminate bacterial endophytic up to 24 days with 0% contamination. | [49] |
| Antibiotic, cefotaxime at 62.5 mg L$^{-1}$ was supplemented to $\frac{1}{2}$ Murashige and Skoog (MS) medium for establishment | Jerusalem artichoke (*Helianthus tuberosus* L.) | It recorded 0% contamination 100% survival of stem nodes cultures. | [45] |
| Copper sulfate (CuSO$_4$ 5H$_2$O) at 60 mg L$^{-1}$ was added to MS medium +3 mg L$^{-1}$ BA + 1 mg L$^{-1}$ KIN for shoot multiplication | Banana (*Musa* sp.) | The growth of the endophytic bacteria was inhibited by recording 0% contamination. | [46] |
| Copper sulfate (CuSO$_4$ 5H$_2$O) at 70 mg L$^{-1}$ was supplemented to MS medium + 5 mg L$^{-1}$ BA for shoot multiplication | *Philodendron selloum* | It eliminated the endogenous bacteria contamination to 0%, without decline in growth of *in vitro* shoots. | [44] |

Abbreviations: Benzylaminopurine (BAP), Murashige and Skoog (MS), benzyladenine (BA).

3.1.2. Delay of Subculture and Burned Plantlets

The *in vitro* micropropagation technique has a lot of benefits; the most important one is represented in producing true-to-type plantlets, which are genetically and physiologically uniform. These plant materials can be used in the intensive production of several plants such as cucumber (*Cucumis sativus* L.) [24], olive trees (*Olea europaea* L.) [49], strawberry (*Fragaria* × *ananassa* Duchesne) [50], cardamom (*Elettaria cardamomum* Maton) [51], and *Flemingia macrophylla* (Willd.) Merr [52]. *In vitro* propagation protocol includes certain stages that should be followed (1) pre-establishment stage, (2) establishment stage (3) multiplication stage, (4) rooting stage, and (5) acclimatization stage, as reported in several articles [47,48,53–56]. However, under heavy work and a lot of tasks in PTC laboratories, the lack of facilities, and a limited number of expert workers and technical specialists, a delay of subculture definitely happens (Figure 3). This delay will for sure cause a great loss or damage in the production of *in vitro* plantlets and even in the maintenance of stock cultures. This situation may differ in the laboratories of developing and developed countries, as it can be clearly observed and spread in the first case for the reasons mentioned earlier. Generally, the effect of time of subculture on the shoot-proliferation rate of *in vitro* cultures varyies from one species to another [57]. In this regard, the long-term incubation period on a culture medium of constant hormonal composition had a negative effect on the multiplication rate of six ornamental species and cultivars of the Rosaceae family [58] and two cultivars of *Dasiphora fruticose* (L.) *Rydb.* [59], but it was found that the longer incubation period (75 days) of pineapple (*Ananas comosus* L. Merr.) *in vitro* shoots on a culture medium resulted in a higher multiplication rate and total number of shoots than a shorter one (30 days) [60]. Somaclonal variation, which resulted due to prolong incubation period in the culture medium (delay of subculture), hinders supplying clonally identical plantlets, which is considered the main target of plant micropropagation [61,62]. The effect of subculture

times on genetic fidelity of *Tetrastigma hemsleyanum* Diels and Gilg callus cultures under a long-term tissue culture was studied [61]. The obtained results clearly indicated that the frequency of somaclonal variation has been increased by increasing the subculture time. Moreover, long-term cultures during clonal multiplication of *Moringa oleifera* shoots resulted in high somaclonal variation [62]. It was proven that supplementation of a culture medium with 50 μM salicylic acid (SA), an anti-ethylene compound, decreased hyperhydricity as well as somaclonal variation under a long-term culture. So, SA was recommended for moringa clonal micropropagation [62]. Burned plantlets are a very common phenomenon during *in vitro* handling, which results from using hot planting tools (mainly forceps and scalpels) during transfer the plantlets (Figure 4). This problem can be avoided when the workers in the PTC laboratory are trained and have enough experience in this field.

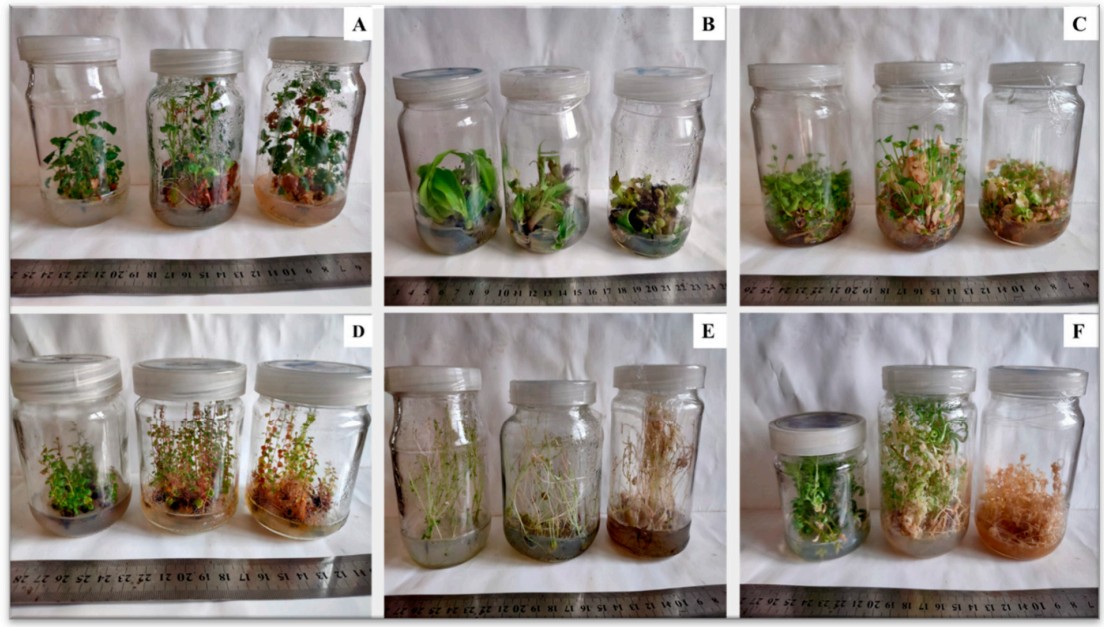

**Figure 3.** Effect of subculture delay on *in vitro* growth and shoot multiplication (in each photo, left jar = normal plant, middle jar = plant in the first stage of degradation, right jar = dead plant). Blackberry (*Rubus ulmifolius* Schott) (**A**), banana (Musa sp.) (**B**), *Philodendron selloum* (**C**), blueberry (*Vaccinium corymbosum* L.) (**D**), potato (*Solanum tuberosum* L.) (**E**), and *Gypsophila paniculata* (**F**).

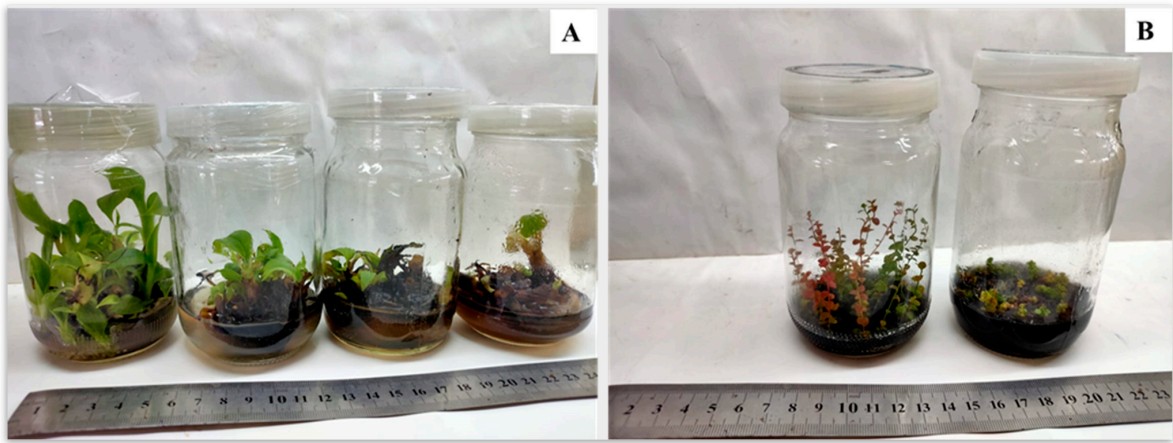

**Figure 4.** Effect of too-high temperature forceps and scalpels during sterilization on *in vitro* shoot multiplication and growth of burned plants. Photo (**A**) represents banana plant (*Musa* sp.) and (**B**) blueberry plant (*Vaccinium corymbosum* L.).

### *3.2. Problems Originated Due to Physiological Reasons*

3.2.1. Browning of Plant-Tissue Cultures

The browning of explants or phenolic browning is a phenomenon that results naturally from enzymatic oxidation of the polyphenolic compounds, which are well-known inhibitors in PTC (Figure 5). Phenols released from injured or cut explants are then oxidized to quinones by polyphenol oxidases (PPOs) and peroxidase (POD), causing browning of the tissues and medium as well [29]. These quinones bind with cell proteins or polymerizes by dehydration, causing the disruption of cell metabolism, inhibition of growth, and, ultimately, death of explants [63]. Browning of tissues could be reduced by timing the explants' collection, supplementing the culture medium with antioxidants, such as citric acid, ascorbic acid, activated charcoal (AC), and polyvinyl pyrrolidine (PVP), alone or in combination, or using liquid culture or micrografting [64–66]. In addition, this phenomenon can be overcome by decreasing the biosynthesis of phenolic compounds and by inhibiting the activity of the phenylalanine ammonia lyase enzyme during the *in vitro* propagation. Moreover, 2-aminoindane 2-phosphonic acid (AIP), an inhibitor polyphenol production, could be added to the culture medium to inhibit oxidative browning [48]. Nitric oxide (NO) was applied to the growth medium to reduce callus browning, which allowed the tissues to recover and regenerate [22]. A comparative study was done among three browning inhibitors that were supplemented to the calli induction medium of bamboo (*Dendrocalamus sinicus* L.C. Chia and J.L. Sun) to prevent browning [67]. These browning inhibitors were sterilized and added to the medium at different concentrations: citric acid (200–600 mg $L^{-1}$), vitamin C (100–300 mg $L^{-1}$), and AC (400–1200 mg $L^{-1}$). Citric acid ($C_6H_8O_7$) at 400 mg $L^{-1}$ was the best inhibitor, significantly inhibiting and reducing callus browning to 17.59%.

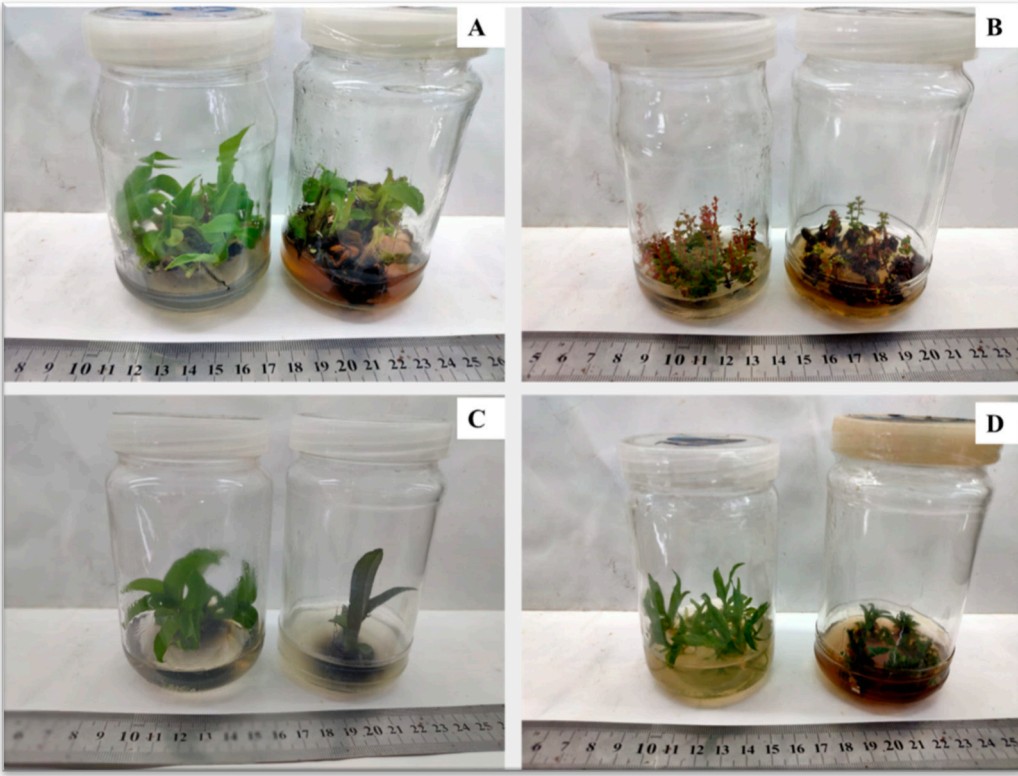

**Figure 5.** The symptoms of browning at the stage of shoot multiplication of *in vitro* plantlets. Left jar = normal plant, right jar = browned plant. Banana plant (*Musa* sp.) (**A**), blueberry plant (*Vaccinium corymbosum* L.) (**B**), *Phalaenopsis* sp. (**C**), and *Cattleya* sp. (**D**).

### 3.2.2. *In Vitro* Rooting Difficulty and Failure of Subsequent Acclimatization

A successful micropropagation protocol requires the appropriate conditions for *in vitro* root initiation and the development of regenerated shoots. Healthy, rooted *in vitro* micro shoots could be acclimatized successfully *in vitro* and/or *ex vitro*; when they are transferred to *ex vitro* conditions, then they can be established in the soil. Previously, the literature investigated the post effect of cytokinins in shoot-proliferation medium on subsequent rooting ability of micro shoots. It was observed that the type and concentration of cytokinin added to the shoot-multiplication medium highly affects the subsequent rooting stage, where rooting frequency has been reduced in many species by supplementation of thidi-azuron (TDZ) or benzyladenine (BA) to the shoot-multiplication medium [68,69], while it strongly inhibited rooting in other species. Moreover, by increasing TDZ concentration in the last shoot-multiplication medium, the number of roots per shoot decreased until it reached 0% at 0.45 $\mu$M TDZ [70]. Similarly, the subsequent rooting of 'Royal Gala' apple regenerated shoots was inhibited at a high concentration of meta-topolin-riboside (TOPR, 6.5 mg L$^{-1}$) in the regeneration medium. Although BA, the most frequently applied cytokinin to the proliferation medium for shoot multiplication, has an inhibitory effect on the subsequent rooting. It was reported that adding BA to the proliferation medium resulted in the decreased rooting ability of 'Red Fuji' apple micro shoots compared to meta-topolin (TOP) [68].

Reducing the salt strength to a half or a third in the rooting medium has been proven to be useful for enhancing root induction, development, and number as well in many leguminous species, fruits, medicinal plants, and trees [65,69,71], while a full-strength MS medium has been widely used for *in vitro* rooting in most herbaceous non-shrub plants that do not have problem in rooting. In this regard, *in vitro* growth of *Typhonium flagelliforme* L., a valuable medicinal herb, is improved by increasing MS medium strength, while the maximum number of roots was recorded with full-strength MS rooting medium [72]. Moreover, *in vitro* regenerated shoots of *Cordyline fruticosa* and *Philodendron selloum* were successfully rooted by 100% after four weeks on full MS medium free-growth regulators [44]. Unlike plants known to be hard-rooting species, it is better to reduce the strength of medium to half, to decrease the osmotic pressure [68]. From that, micro shoots of sweet potato (*Ipomoea batatas* (L.) Lam. were *in vitro* rooted with 100% rooting after three weeks culturing on half-strength MS medium [73]. It was proven that the optimal medium for *in vitro* rooting of *Arnica montana* L. and Cattleya orchids hybrid was half-strength MS supplemented by 0.5 and 2.5 mg L$^{-1}$ indole 3 butyric acid (IBA), respectively [71,74]. It is widely accepted that auxin is needed for root induction, but its absence or continued presence in the medium inhibits the adventitious roots of apple micro shoots [65]. Moreover, root quality may be improved by addition of auxins [69].

In addition, the type of auxin added to the rooting medium is mainly dependent on several factors; the most important one is the protocol used for *in vitro* rooting, if it is one-phase or two-phase, while the latter one is widely applied for most horticultural plants such as apple (*Malus domestica* Borkh.). It was reported that the rooting of micropropagated apple shoots includes a short phase of one week for root induction, in which IBA was frequently applied, followed by a longer phase of several weeks where naphthaleneacetic acid (NAA) was the most-common auxin supplied to the medium for root elongation [65,66]. Moreover, the optimal auxin for rooting was plant-species-dependent. Therefore, the choice of auxin is a very important issue for woody plants, which show difficulty in *in vitro* rooting. It was found that IBA has been commonly used, rather than indole-3-acetic acid (IAA) and NAA, as the most effective auxin to induce root formation in woody species such as apple and *Cassia angustifolia* Vahl.; whereas, 1, 2-benzisoxazole-3-acetic acid (BOA) showed very poor effect on root induction in woody plants [66]. In this regard, IBA was the most efficient among the other auxins tested for root induction from the *in vitro* shoots of *Cymbidium aloifolium*, *Dendrobuim aphyllum*, *Dendrobuim moschatum*, and Cattleya hybrid [74,75]. Moreover, the IBA applied method may be effective on *in vitro* root initiation. It was recorded that shoots could be cultured directly on rooting medium supplemented

with IBA or dipped in IBA-filter sterilized solutions, then cultured on MS medium for rooting induction. It was stated that chickpea *(Cicer arietinum* L.) 'Giza4' *in vitro* shoots had the highest rooting potential, rooting percentage, number of roots, and root length, when shoots were dipped in 50 mM L$^{-1}$ IBA and cultured in liquid medium [76].

On the other hand, auxin concentrations in rooting medium influence the rooting ability in many species. Low auxin concentrations were optimal for improving the rooting of legumes for root development forming adventitious roots, and it increased the number of roots per shoot. Furthermore, the carbohydrate source and its concentration can affect the *in vitro* rooting in many species where sugar is essential for root formation, by supplying the micro shoots with energy. No roots could be developed in apple 'Merton Malling' rootstocks on a sugar-free medium [65]. It was demonstrated that a low sucrose concentration (20 mg L$^{-1}$) is preferable to enhance the rooting of *Astragalus chrysochlorus* to reach 93% [77], whereas 1% (10 mg L$^{-1}$) sucrose was the optimal concentration for rooting of the 'Granny Smith' apple in liquid culture medium containing 10 μM IBA. There are other factors that can impact the *in vitro* rooting of micro shoots [65], such as cultivars or genotypes, because they are different in their rooting ability and, thus, their optimal rooting medium. The orientation of micro shoots was reported to be a factor altering the efficiency of *in vitro* rooting. It was reported that the rooting percentage of apple micro shoots significantly depended on the interaction between the micro shoots' orientation, if it is upright or inverted in rooting medium, genotype, and auxin level [78]. Moreover, the number of subcultures and shoot orientation for the last shoot-multiplication medium prior to the rooting stage could influence the efficiency of the subsequent rooting process. The best result for rooting percentage (≈89%) was recorded for apple micro shoots placed horizontally on the last medium for shoot multiplication one week before transferring on rooting medium [79]. It was observed that rooting could be improved in several cultivars by increasing the number of subcultures [65]. Activated charcoal as organic compound could be employed to enhance root length, if it is added to root elongation medium. These previous factors should be taken in consideration during *in vitro* rooting, especially for difficult-to-root plant species as woody plants.

Due to many reasons, such as low humidity, high level of irradiance, low root uptake of water under *in vitro* conditions [80], rapid desiccation of *in vitro* plantlets, and their easy infection by fungal and bacterial diseases; *in vitro* rooted plants might fail when they are transferred to *ex vitro* conditions during acclimatization. To overcome those obstacles, *in vitro* plants should be supported gradually by following the right acclimatization strategy and right selection of the acclimatization medium [81]. Thus, the ultimate success of the *in vitro* micropropagation of plants depends on the successful establishment of plantlets in the soil through the acclimatization process [82]. Application of biostimulators was proven to be used for successful acclimatization. The fungus of *Piriformospora indica* has been used as a biological stimulator to enhance plant growth and root development during the acclimatization process, allowing *in vitro* hard rooting plants such as orchid to be transferred to the field [83]. Another plant-growth-promoting microorganism (*Pseudomonas oryzihabitans*) has been applied to the *in vitro* rooted plantlets of pear to boost the growth efficiency in soil [84]. The physiological aspects of *in vitro* rooted banana plants (*Musa* sp.) have been improved through inoculation of *Buttiauxella agrestis* and *Bacillus thuringiensis* (plant growth-promoting bacteria that produce auxins) into the root system of banana at the acclimatization stage. Hence, high quality seedlings have been acquired [85]. In addition, many supporting materials (e.g., agar, rockwool, perlite, vermiculite) have been used to improve the growth and survival rates of *in vitro* plants in *ex vitro* conditions [86].

### 3.3. Problems Originated Due to Genetical Reasons
Somaclonal Variation

The somaclonal variation point to culture-induced, unexpected, and undesired variations or anomalies, which eventually is inherited by the clonal progenies, becomes genetic and happens during tissue cultures; this is, thus, noticed in the *in vitro* regenerated plants,

and is an often an unwanted phenomenon, especially in mass micropropagation programs that are highly desirable for the production of plant material of the same type as the mother (genetically uniform) by maintaining the genetic stability of the *in vitro* regenerated plantlets [24,25]. Somaclonal variation is genotype-dependent, where some genotypes or species are more amenable to these changes than others when they are cultured *in vitro* [87]. There are other reasons for genetic variability, including the period of the *in vitro* culture of the explant [88], composition of the culture medium in particular auxins and cytokinins as plant-growth regulators, type of sugar [87], proliferation rate of tissues, interval between subcultures, natural selection, mutations in *in vitro* cultures [29], ploidy level, and mode of regeneration used. *In vitro* culture of the explant causes stress on the plant cells and makes them undergo a genomic shock or change [25]. Furthermore, mechanical factors, such as damage of explant or its exposure to sterilizing substances, as well as instability in temperature, humidity, and lighting, can promote somaclonal variation [89].

Many molecular studies on somaclonal variation have been published on different crops, such as orchid (*Dendrobium* sp.) [90], olive (*Olea europaea* L.) [91], date palm (*Phoenix dactylifera* L.) [92], cucumber (*Cucumis sativus* L.) [24], pineapple (*Ananas comosus* L.) [93], and flax (*Linum usitatissimum*) [94]. The great potential of uses of somaclonal variations in agriculture may include producing new generations of new agronomic variants with favorable traits and generations of disease resistance varieties. Somaclonal variation in different plant species can be assessed by biochemical, morphological, and molecular-based markers methods [25]. The main problems of somaclonal variations may include (1) generation of unpredictable and uncontrollable variations particularly non-agricultural use. (2), generation of somaclonal variants that is limited only to those plants that have ability to multiply in culture medium and regenerate into whole plants, (3) "*somaclones may be having low growth rate as well as reduced fertility*" [25], and (4) somaclonal variations prevent obtaining true to type plants, which is considered a main issue in micropropagation protocols. To overcome somaclonal variation, we have to avoid long-term *in vitro* cultures [61,62] and overexposure to phytohormones in culture medium as well as regularly reinitiate clones from new explants, plus the number of subcultures should be kept at a minimum and the use of 2,4-dichlorophenoxyacetic acid (2,4-D) and kinetin (KIN) in tissue cultures should be rationalized, because they are known to induce variations [95].

## 4. Disorders and Abnormalities in Plant Micropropagation

PTCs are considered as an important tool used in the micropropagation, conservation, and genetic improvement of plant species in breeding programs. However, such applications, in particular the micropropagation technique, are hindered by some disorders and abnormalities that will be discussed in the following subsections.

### 4.1. Hyperhydricity

The term of hyperhydricity (HH) mainly describes a physiological disorder or malformation resulting from the immoderate hydration, little lignification, and, thereby, diminished mechanical strength of *in vitro* cultured plants [96]. It occurs in herbaceous, woody plants and succulent plants. This term is also called glassiness, glauciness, translucency, and vitrescence, and it used to describe this physiological disorder (Figure 6). The hyperhydrated or vitrified *in vitro* plants appear to be turgid or hyperhydric and hypo-lignified, and their surfaces are watery. Somehow, their organs are translucent, less green in some cases, and they break easily [21]. HH seriously affects the regeneration and micropropagation of plants, causing considerable commercial losses in *in vitro* propagation industry by decreasing the multiplication rate and the quality of the *in vitro* cultured plants. It also limits the use of tissue-culture techniques in plant-resources conservation as well as genetic improvement by transformation [97].

The main reasons for the HH in PTCs may include high relative humidity, high salt concentration of the medium, low light intensity, gas accumulation in the jar, length of time intervals between subcultures where a lengthy subculture enhanced ethylene accumulation

in culture vessels that led to HH [98], number of subcultures where upon increasing the number of subcultures of *in vitro* cultured plants the occurrence of HH increases [99], low calcium content, and high ammonium content, which may trigger oxidative stress [100–102]. There are different factors that impact HH, which can be classified as follows: (1) explant, including physiological age, genotype, organ type, and size; (2) media components, such as basal medium, gelling agents, and plant-growth regulators; (3) culture conditions, such as ventilation and light intensity; and (4) exogenous additive substances, such as salicylic acid (SA), polyethylene glycol (PEG 6000), hydrogene peroxide ($H_2O_2$), and $Ag^+$ [103]. This phenomenon could be controlled, as reported in some studies, by supplementing the culture medium with $AgNO_3$ [74] or ascorbic acid, salicylic acid, spermidine, and $H_2O_2$ [103]. Gelling agents are one of the components in the culture medium, and the type and concentration of them significantly affects the quality of the *in vitro* plant, as they cause HH [104]. So, HH could be prevented in shoots by replacing gelrite with one of the other gelling agents such as Agar-Agar, Danish Agar, or Cero Agar Type 8952 at 7 g $L^{-1}$ [100,105], as they have a high content of P, Ca, Na, B, and Cu. It was noticed that using gelrite as a gelling agent caused a lower multiplication rate and almost four-fold HH (65%) in *Aloe polyphylla* and two times higher HH in teak (*Tectona grandis* L.) [106], higher than that in an agar-solidified medium, due to its physical properties that lead to a higher water percentage in *in vitro* cultured tissues, as well as because of the small amount of gelrite put in the medium; thus, the quantity of water is greatly causing HH [107]. Moreover, this phenomenon could be minimized by ensuring adequate gas exchange with ventilation and light-emitting diodes facilitation [101] or supplementation of the growth media with polyamines (putrescine, spermidine, or spermine) [108] (Table 3). More studies have been carried out on the HH of many plants such as *Musa* spp. [53], *Dianthus chinensis* L. [96,108], *Cycladenia humilis* var. jonesii [109], and *Dendrobium officinale* [110].

**Table 3.** Some substances used to control hyperhydricity in plant-tissue cultures.

| Plant Species/Cultivar | Applied Substance (Name and Concentration) | Success (%) | Refs. |
|---|---|---|---|
| Pink (*Dianthus chinensis* L.) | $AgNO_3$ at 29.4 µM $L^{-1}$ supplemented to culture medium | 67% of the hyperhydric *Dianthus chinensis* L. *in vitro* plantlets have been revert to normal appearance. | [97] |
| Garlic (*Allium sativum* L.) | Ascorbic acid at 250 µM + salicylic acid at 50 µM + spermidine at 10 µM + $H_2O_2$ at 50 µM | Hyperhydricity was relieved. | [103] |
| Olive (*Olea europaea* L.) cv. 'Gemlik' | Agar-agar at 0.65% *w/v* supplemented to culture medium | Hyperhydricity was prevented by changing the gelling agent from gelrite to Agar-Agar. | [100] |
| China pink (*Dianthus chinensis* L.) | Polyamines as spermine at 5 µM supplemented to MS medium | It significantly reduced the hyperhydrated shoots to 0.33% compared to control treatment (100%), and the number of healthy reverted shoots was maximum (11.0). | [108] |

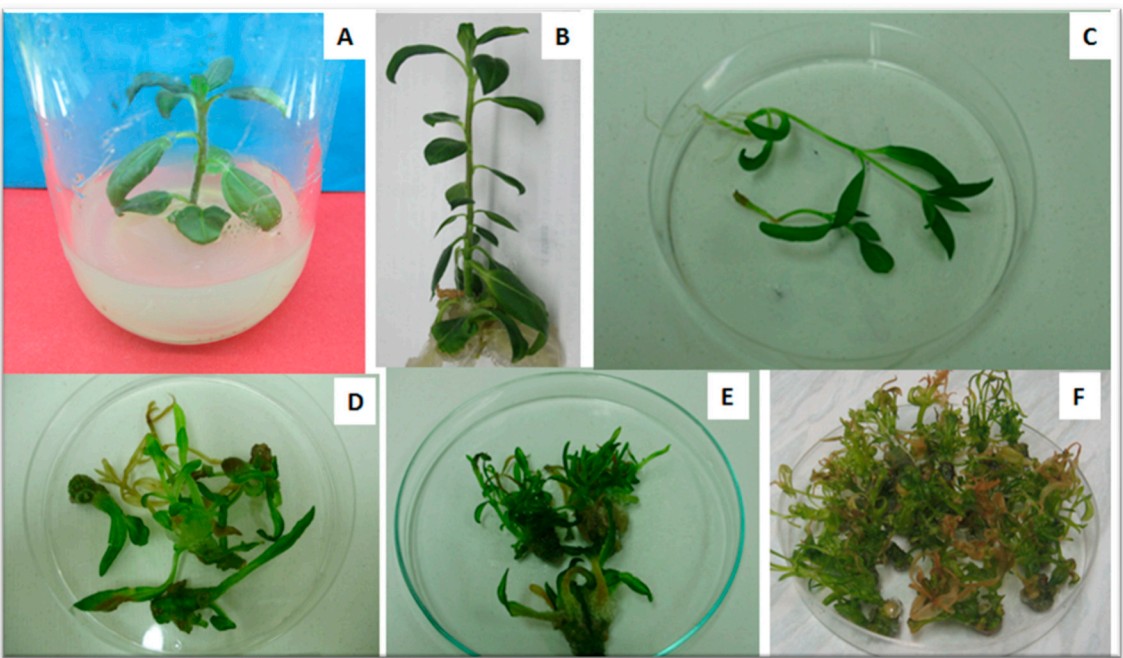

**Figure 6.** Effect of hyperhydricity on *in vitro* shoot proliferation of Jerusalem artichoke plant (*Helianthus tuberosus* L.). Photos (**A–C**) = normal *in vitro* plantlets, whereas photos (**D–F**) = hyperhydrated *in vitro* plantlets.

### 4.2. Shoot Tip Necrosis

Shoot-tip necrosis (STN) is defined as a physiological status that leads to death of the shoot tip of *in vitro* shoots and is caused by the culture conditions [23]. Moreover, STN results in shoot tip burn, injury, or dieback [23]. STN may occur when the shoot tip of a plantlet has some abnormal features, such as in the elongation and/or rooting stages or browning and death during multiplication, although the growing of plantlets may be appeared as ideal conditions for both *in vitro* and *ex vitro* cases. STN also could be displayed in several plants, particularly woody shrubs and trees at 21.9% and 58.9%, respectively [23]. The precise mechanism of STN still needs more clarification, but some reasons have been proposed such as the deficiency of some minerals (e.g., Ca, Mg, K, B, or $NO_3^-$), an imbalance of nutrients, accumulations of phenolic compounds in shoot tip, a lack of antioxidants, a low level of $Ca^{2+}$, the presence of high concentrations of PGRs in the culture medium, or the kind and concentration of cytokinins in the medium [111]. STN may cause severe losses in micropropagation of many tree species such as apple (*Malus domestica* Borkh.) [112], grape (*Vitis* sp.) [113], pistachio (*Pistacia vera* L.) [114], *Prunus yedoensis* [115], and *Prunus yedoensis* [116]. Several methods have been employed to relieve STN, i.e., the supplementation of 50–100 mg $L^{-1}$ calcium chloride to MS medium, which allows recovery of 90% of banana and plantains (*Musa* spp.) shoots [117], or the addition of 0.50 mM fructose + 1.0 mM calcium chloride to culture medium, which gave 100% success to control STN in plum (*Prunus salicina* L.) [116]. Adding adenine sulphate at 100 mg $L^{-1}$ with 10 µM BA was the most effective to inhibit STN in 90% of *Syzygium cumini* L. cultures [118]. Applying ethylene-inhibitor compounds to the shoot-regeneration medium, such as Aminoethoxy vinyl glycine (AVG; 12.5 µM AVG), silver thiosulfate (STS; 1 µM), and sodium nitroprusside (SNP; 20 µM), has been proven to enhance apical shoot initiation and multiplication rates as well as reduce the leaf senescence (yellowing) and shoot necrosis of *in vitro* roses through ethylene biosynthesis inhibition [119].

### 4.3. Albino Plantlets (Albinism) in Anther Cultures

Albinism is referring to incomplete differentiation of chloroplast membranes, and the subsequent total or partial loss of the chlorophyll pigments of *in vitro* plants [120]. It can

negatively affect the efficiency of the photosynthesis process in plants, which may reduce their survival (Figure 7). Moreover, changes in the biochemical parameters of green and albino leaves of *in vitro* cultures of *Caladium bicolor* were observed [121]. Furthermore, the albinism of *Agave angustifolia* is due to molecular alterations in the leaves' meristematic cells of albino plants, leading to morphological abnormalities that affect stomatal conductance, transpiration, and photosynthesis, while they cause the loss of the physiological functions and the development of the stomata [122].

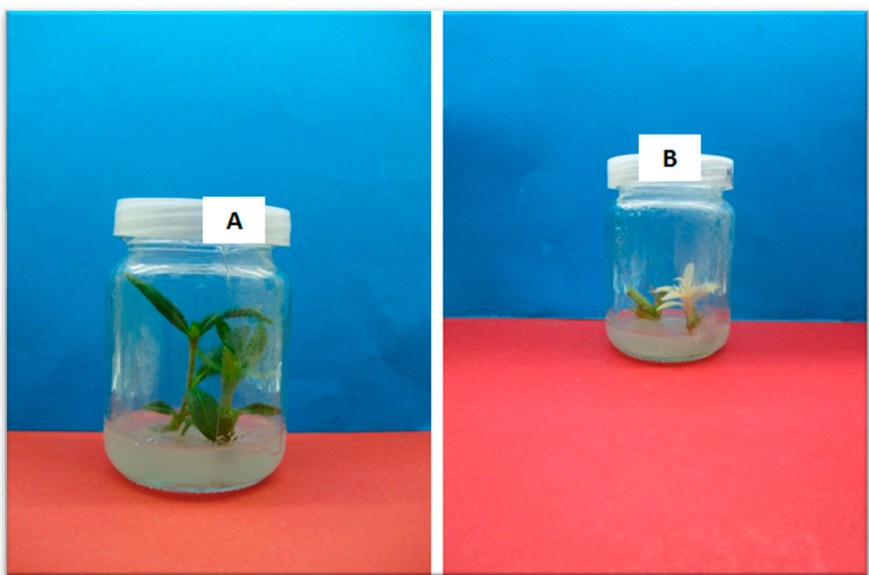

**Figure 7.** Albino phenomenon of *in vitro* propagated Jerusalem artichoke plant (*Helianthus tuberosus* L.). Photo (**A**) = normal plantlets, whereas photo (**B**) = albino plantlets.

The anther culture technique (androgenesis), which has been widely and successfully applied to stimulate haploid or doubled haploid plants of wheat and other species, such as *Datura*, is associated with a high frequency of albino seedlings that limits its usage in breeding programs. Thus, albinism phenomenon is still a critical obstacle for cereal-improvement programs. Although the breeding protocol based on *in vitro* anther cultures compared to traditional ones has many advantages, as it decreases breeding time, accelerates homozygositym and improves selection efficiency, the high albino rate is its most-limiting factor because the albino seedlings fail to grow and develop into adult plants [123]. The high albinism rate has been reported in anther cultures of other cereal plants, such as rice (*Oryza sativa* L.) [124], barley (*Hordeum vulgare* L.) [125], and maize (*Zea mays* L.) [126]. The early investigations stated that the cause of albinism is mainly due to variation in plastid genes, where the heterogeneous population of chloroplast genomic molecules has been noticed in most albino plants. The plastid genome exists in a linear form in some albino wheat and barley plantlets [127]. Other studies have proven that albinism is due to a genetic variation in nuclear genes, through an alteration in the nuclear genes that are encoding the proteins of chloroplast in albino plants [128]. In many $C_4$ grasses such as sugarcane (*Saccharum* spp.), albinism appears because of a heritable genetic mutation. Current research on oat (*Avena sativa* L.), barley (*Hordeum vulgare* L.), and wheat (*Triticum aestivum* L.) suggests that the albino chlorosis is due to genetic mutations that impacts plastid production, function and/or carotenoid synthesis, or it may be due to photooxidative damage occur to chloroplasts [129]. Recently, it was found that albinism in wheat is linked to the transcriptional suppression of the specific genes responsible for chloroplast biosynthesis during the first weeks of *in vitro* anther culture [130].

Adding copper sulfate at 10 μM to anther pre-medium improved the regeneration of green doubled haploid plants from cultivars known to produce mainly albino plants, when the ratio of albino plants is reduced. It was proven that copper; an essential element for

chlorophyll synthesis; increased the plastid density in microspores by divisions and enhanced the development of microspore proplastids into chloroplasts [131]. The pre-medium is liquid medium of pH 5.8, containing demineralized water and 62 g L$^{-1}$ mannitol, and its osmotic pressure is 180 mosmol L$^{-1}$, when barley anthers were kept in the dark at 4 °C and 80% relative humidity for 4 days for cold pre-treatment or stress-inducing pre-treatment. This cold treatment is reported to stop the gametophytic development of microspores and enhance the sporophytic developmental pathway, where microspores develop into embryos that can further develop into haploid or spontaneously doubled haploid plants. Moreover, albinism has been overcome in horse chestnut (*Aesculus hyppocastanum* L.) by supplementation of androgenic embryos' medium with 0.01 mg L$^{-1}$ abscisic acid [132]. Moreover, understanding the molecular background of the albinism phenomenon may help to develop efficient methods to minimize the albino rates in anther cultures [123]. On the other hand, this phenomenon is desirable in other plants such as ornamental plants, which will be more attractive if they have white or albino organs. Moreover, albino mutants could be useful for chlorophyll biosynthesis research and cloning the associated genes in plants [123].

*4.4. Recalcitrance in Clonal Micropropagation*

Recalcitrance refers to failure or disobedience of cells or tissues of the explants to response to a regenerative condition, even by somatic embryogenesis or organogenesis (Figure 8). It is defined as disability of plant cells, tissues, and organs to respond to *in vitro* culture manipulations [133]. Recalcitrance is still a major obstacle in the clonal propagation of many plant species, in particular in their mature phase of development as adult conifers [134]. It is well-known that juvenile tissues show higher morphogenetic ability than adult ones, so *in vitro* cloning of adult specimens is still problematic due to recalcitrance phenomenon. In other words, there are some tissues that are more amenable to express totipotency than others in the plant body. So, *in vitro* recalcitrance could be avoided by selecting convenient explants at the suitable developmental responsive stage to be able to regenerate, directly or indirectly, via organogenesis or somatic embryogenesis, respectively [135]. Another study suggests that controlling expression of genes involved in embryo initiation could be a promising approach to overcome recalcitrance [134]. Thin-cell-layers (TCLs) technology (excising thin and transverse or longitudinal slices of tissues then cultured *in vitro*) or protocorm-like body (PLB) allows isolation of totipotent tissues that may be exposed to restrictions from their neighboring inhibitory tissues. This method has been applied successfully for the clonal micropropagation of many recalcitrant herbaceous and tree species or orchids [136,137]. These PLBs are considered to be somatic embryos, so could be used for large-scale mass propagation in bioreactors or in synthetic seeds for long-term storage [136]. In this regard, an effective *in vitro* regeneration protocol via somatic embryogenesis was established from the stem transverse thin-cell layers (tTCLs) of an orchid (*Dendrobium aqueum* (Lindl.) Kuntze) [138].

The genus *Capsicum* has been reported as recalcitrant toward *in vitro* propagation, when formation of rosette leafy structures makes elongation of regenerated shoots very difficult. The regenerated shoot buds of *Capsicum annuum* L. could be elongated to be 4.5 cm on shoot elongation medium (MS medium enriched with 0.87 µM gibberellic acid (GA$_3$) and 39.64 µM phloroglucinol; PG) [139]. The mode of action of PG was through inhibiting auxin oxidation and reducing the vitrification, thus, stimulating shoot elongation, roots growth, and development. Chen et al. [140] reported that the elongation of regenerated shoots from anther cultures of flax (*Linum usitatissimum* L.) has become the limiting factor for further improvement and production of doubled haploid plants that are needed by breeders to develop new varieties in a short time. They proved that MS medium supplemented with a low content of sucrose (10 g L$^{-1}$) was the optimal medium for shoot elongation. This study suggested that sucrose could act as a carbon source and an osmotic regulator as well. The negative effect of the high concentrations of sucrose on shoot elongation indicated that the osmotic pressure of culture medium suppresses further shoot development. The efficiency

of TDZ was stated for *in vitro* propagation of many plants by reducing the recalcitrant nature of these species [141]. TDZ enhances morphogenesis by stimulation and modulation of the endogenous plant-growth regulators in the cultured tissues.

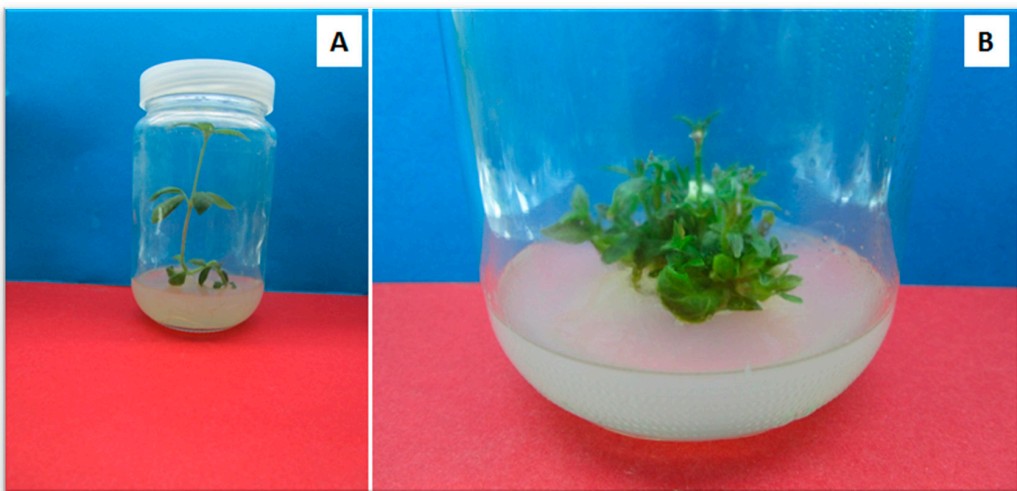

**Figure 8.** Rosette phenomenon (recalcitrant to elongation) of *in vitro* Jerusalem artichoke plant (*Helianthus tuberosus* L.). Photo (**A**) = normal elongated shootlet, whereas photo (**B**) = rosette shootlet.

*4.5. Shoot Abnormalities*

Shoot distortions are still limitations that impede the use of micropropagation as an alternative and rapid technique to produce huge numbers of genetically uniform and pathogen-free plantlets. Formation of basal callus during the shooting stage could inhibit normal axillary-shoot proliferation and direct the regeneration pathway to callogenesis and indirect organogenesis. This phenomenon (basal callus) is attributed to the auxins that are accumulated in the basal ends of shoots and stimulate cell division, forming the callus (Figure 9). Thidiazuron (TDZ) could induce the callus base on *in vitro* shoots of many plants. TDZ also shows auxin-like activity by modifying the biosynthesis and accumulation of endogenous auxins [141]. Basal-callus formation has been noticed as an undesirable effect of TDZ for many plant species [141]. It was proven that formation of the callus at the basal ends of nodal explants on a medium rich in cytokinin (CK) is common in plant species of strong apical dominance. In addition, a swollen shoot base is considered as one of shoot deformities that negatively affect shoot quality and plantlets. It could result on a medium supplemented with 13.6 µM TDZ, when normal shoots are produced at 0.5 µM TDZ [141]. To avoid shoot abnormalities induced by TDZ, it should be added at very low or pulse concentrations and the period of exposure must be shortened as well.

Moreover, yellowing and leaf abscission of *in vitro* cultured plants is one of the abnormalities affecting the shoots growth and multiplication. This phenomenon is observed for *in vitro* rose due to the accumulation of ethylene gas (maturation and aging hormone) and activity of hydrolysis enzymes such as pectinase and cellulase, which cause degradation in the cell walls of the abscission zone and reduce lignification, so then leaves more easily fall [142]. Silver and cobalt nanoparticles (Ag-NPs and Co-NPs, respectively) have been used to control leaf abscission in rose-tissue cultures by limiting ethylene biosynthesis and inhibiting the enzymatic hydrolyses of pectin and cellulose. It was recorded that Ag-NPs at 2 mg L$^{-1}$ was the optimal for the mass propagation of shoots, which also improved the quality of rose micro shoots, whereas supplementation of 4.65 µg L$^{-1}$ Co-NPs to the rooting medium increased the number of roots and the root length as well as decreased the ethylene gas content, pectinase, and cellulase activity, gave the highest survival rate (96.67%), enhanced growth and development during acclimatization, and increased the quality of *in vitro* roses plantlets [142]. Many studies, aimed to develop an efficient protocol for the rapid propagation of Australian finger lime (*Citrus australasica*), failed due to the

excessive leaf drop caused by increasing the ethylene synthesis and accumulation in culture vessels. However, silver compounds, such as silver thiosulfate (STS) and silver nitrate (Ag-NO$_3$), and silver nanoparticles (Ag-NPs) have been applied to regulate leaf drop by managing ethylene biosynthesis. The obtained results clearly confirmed that silver thiosulfate at 60 µM was the best to control leaf abscission and improve *in vitro* regeneration of finger lime [143].

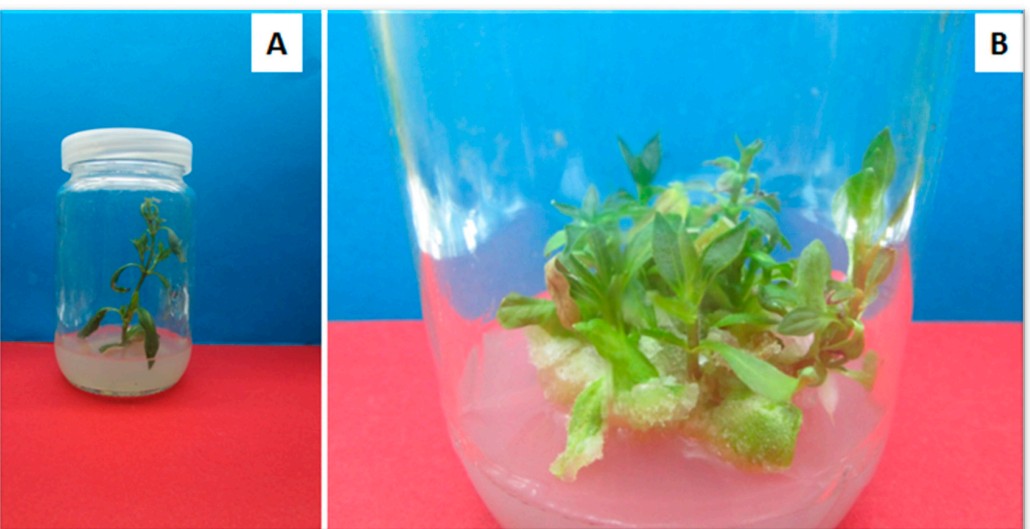

**Figure 9.** Effect of basal callus on *in vitro* shoot proliferation of Jerusalem artichoke plant (*Helianthus tuberosus* L.). Photo (**A**) = normal shootlet, whereas photo (**B**) = shootlets with base callus.

*4.6. In Vitro Habituation*

It is the natural phenomenon, whereby tissue cultures continue to develop in the absence of auxins or cytokinins, because the endogenous biosynthesis by the cultured tissue leads to an autotrophic or independent state (auxins or cytokinins-sufficiency), which is called habituation [144], as shown in Figure 10. These cultures do not need the supplementation of external auxins or cytokinins, when they lose their requirements for these specific hormones [145]. This phenomenon resulted from a prolonged culture period. Early studies explain that cytokinin independence may be due to overproduction by the cultured tissues [146]. Habituation can be transferred to the new cultures, when the callus derived from the habituated ones keeps the cytokinin-independent state. A specific gene (the cytokinin receptor *CRE1*), whose overexpression may responsible for the habituation as a naturally occurring phenomenon, has been identified by Pischke et al. [145]. The definition of habituation as "the ability of plant tissues to grow or regenerate on plant growth regulators-free medium" was recorded by Gautheret for the first time [147]. It was noticed that the habituated rice calli (H-calli) showed decreasing levels of lipids and starch, the two major storage metabolites in higher plants, compared to non-habituated ones (NH-calli). This is attributed to their elevated consumption of lipids and starch in the absence of an exogenous addition of hormones, to ensure their requirements of energy to maintain an optimum growth and development, so they can survive and grow under an exogenous growth regulator free medium. Another explanation for habituation supposes that the metabolism of lipids and starch may be slowed down by habituated rice calli, to adapt with the hormone-free medium [148]. The two types of rice calli (H-calli and NH-calli) were reported to have completely different endogenous hormonal status. The indole-3-acetic acid (IAA) content in habituated calli was seven-fold higher than that in non-habituated ones. On the other hand, there was a decrease in abscisic acid (ABA), ethylene, and gibberellins (GA$_{12}$, GA$_{19}$, GA$_{29}$) in the habituated calli. This explains why the habituated rice calli can grow faster than the non-habituated ones [148].

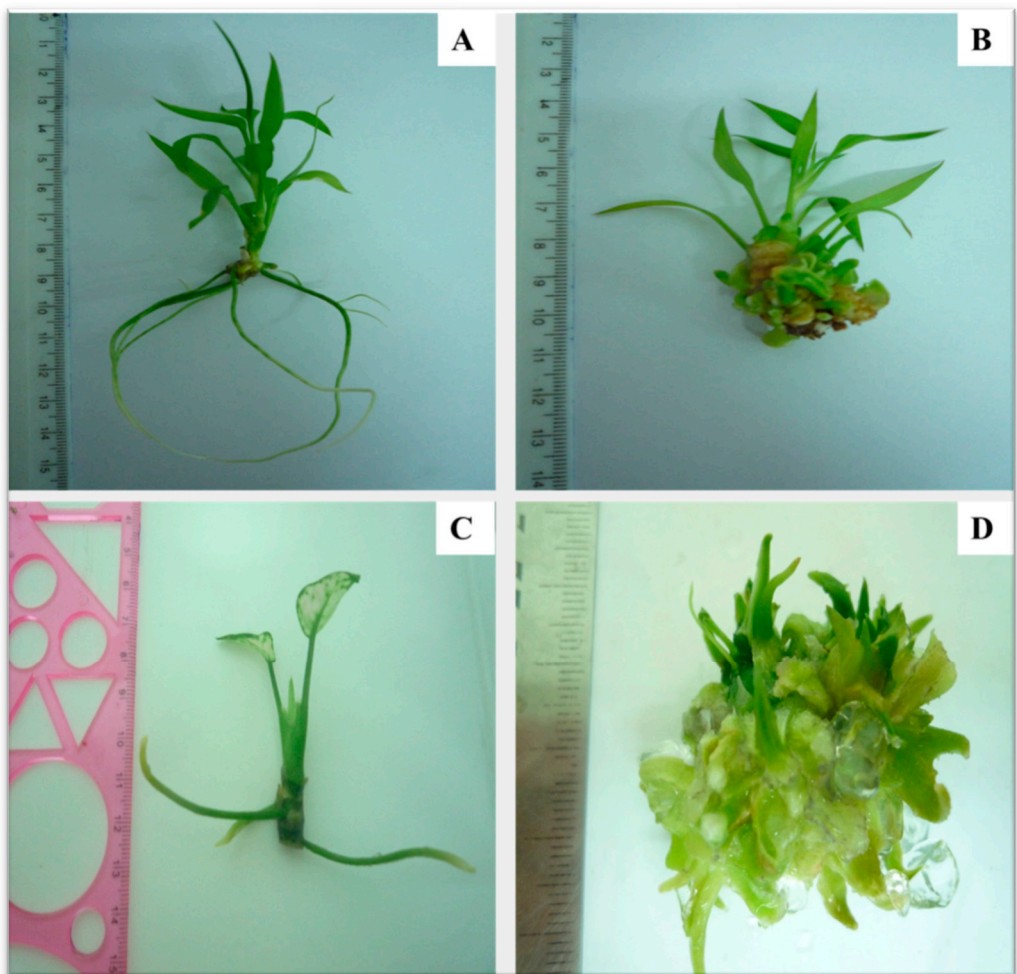

**Figure 10.** Effect of *in vitro* plant habituation on growth and rooting induction. (**A**) Habituated rooted *Cordyline terminalis* plant, (**B**) non-habituated *Cordyline terminalis*, (**C**) habituated rooted *Aglaonema* sp. plant, and (**D**) non-habituated non-rooted f *Aglaonema* sp. plants maintained in the absence of cytokinin for 6 weeks.

## 5. General Discussion

In this section, we need to answer about the main following questions arisen in this article:

Is this study (the current review) an important issue for future research of plant-tissue culture?

What is the main motivation and justification of this work?

What are the major challenges facing the field of plant-tissue culture?

Are these challenges facing both developing and developed countries?

To what extent is this study crucial in the field of PTC?

What are the open questions that still need to be answered?

Concerning the first question, yes, this article may be useful for the field of PTC because it contains many physiological, technical, and genetical problems, which the workers of PTC laboratories face during the micropropagation of plants and the suggested solutions. This article is the first report that includes all these problems together. So, we think this work is a useful for all workers in the field of plant-tissue culture. [32]. Plant micropropagation is one of the optimum biotechnological approaches to overcome many global problems such as desertification, salinization, climate change, and the global water crisis. Applied plant biotechnology via micropropagation is considered a part of the sustainable solution to meet the goal of 'zero hunger' by increasing food security [149,150].

What is the main motivation and justification of this work? Due to some of us working in both the academic field (as a full professor at the university) and the technical field (as a consultant at a company for plant biotechnology), we wanted to invest more than 25 years of expertise to present this work. So, this is the main motivation and justification of this work. This work is useful for the teaching or research purposes of academic staff (students and teachers) and on the commercial level as well. Recently, a great concern about micropropagation and plant-biotechnology research and commercial scale was published in some distinguished books on topics such as agricultural biotechnology [2,9,151], commercial tissue culture in horticulture [152], and plant-tissue culture techniques in tropical horticultural species [153].

What are the major challenges facing the field of plant-tissue culture? Many problems are facing plant micropropagation, one of the most important techniques in plant-tissue culture, as presented in Figure 11. These problems include technical problems (contamination, delay of subculture, burned plantlets), physiological problems (browning of plant-tissue cultures, *in vitro* rooting difficulty, failure of subsequent acclimatization), and genetical problems (mainly somaclonal variations). Disorders and abnormalities in plant micropropagation are sound problems in this field, which may include hyperhydricity, shoot tip necrosis, albino plantlets, recalcitrance in clonal micropropagation, shoot abnormalities, and *in vitro* habituation (Figure 11). In some cases, socioeconomic factors are considered as the major constraints besides the previously mentioned ones, which may include the loss or lack of skilled human resources, inadequate infrastructures, and the high-cost of technology development and its transfer, as reported in South Africa [154]. Contamination is considered the great and most important factor controlling the success of micropropagation in plant-tissue-culture laboratories [29]. Selecting the proper protocol for plant micropropagation is also a crucial factor, which depends on the plant species. For standardization of the micropropagation protocol of any plant species, many studies have been recently reported (e.g., [155–157]).

Are these challenges facing both developing and developed countries? Several challenges face both developing and developed countries, although some problems belong solely to developing countries, which mainly are linked to technical and human facilities. The dissemination of research outputs is a serious challenge facing all countries, by converting the high-quality research outputs of institutions and universities into economic value under the domain of the mega-companies in plant biotechnology. So, it has been stated that "*No single published document has been known yet to provide an overview of recent studies conducted in the field of plant cell, tissue, and organ culture. Especially, reference material about new technologies and practical methods currently applied in research and production is unavailable*" [153].

To what extent is this study crucial in the field of plant micropropagation and PTC? Due to the discussion of the many problems or challenges of the field of plant micropropagation in one article, this work is thought to be useful. Why? Since this review depended on the information from two main sources, the first is from high-impact journals, which have a sound reputation in the field of plant-tissue culture, while the second involves information, for more than 25 years, from both the academic and practical fields of plant biotechnology. So, this study set out with the aim of assessing the importance of problems or challenges in plant micropropagation, using some photos or drawing figures for better understanding and more elucidation.

What are the open questions that still need to be answered? Plant micropropagation has several open questions, which need to be answered in the future. These further questions stem from the main factors that control plant micropropagation and their stages, such as the selection of proper stock plants, establishment of aseptic culture, multiplication of explants, rooting of regenerated shoots or somatic embryo germination, and acclimatization stage or transferring of plantlets into cultivated soil.

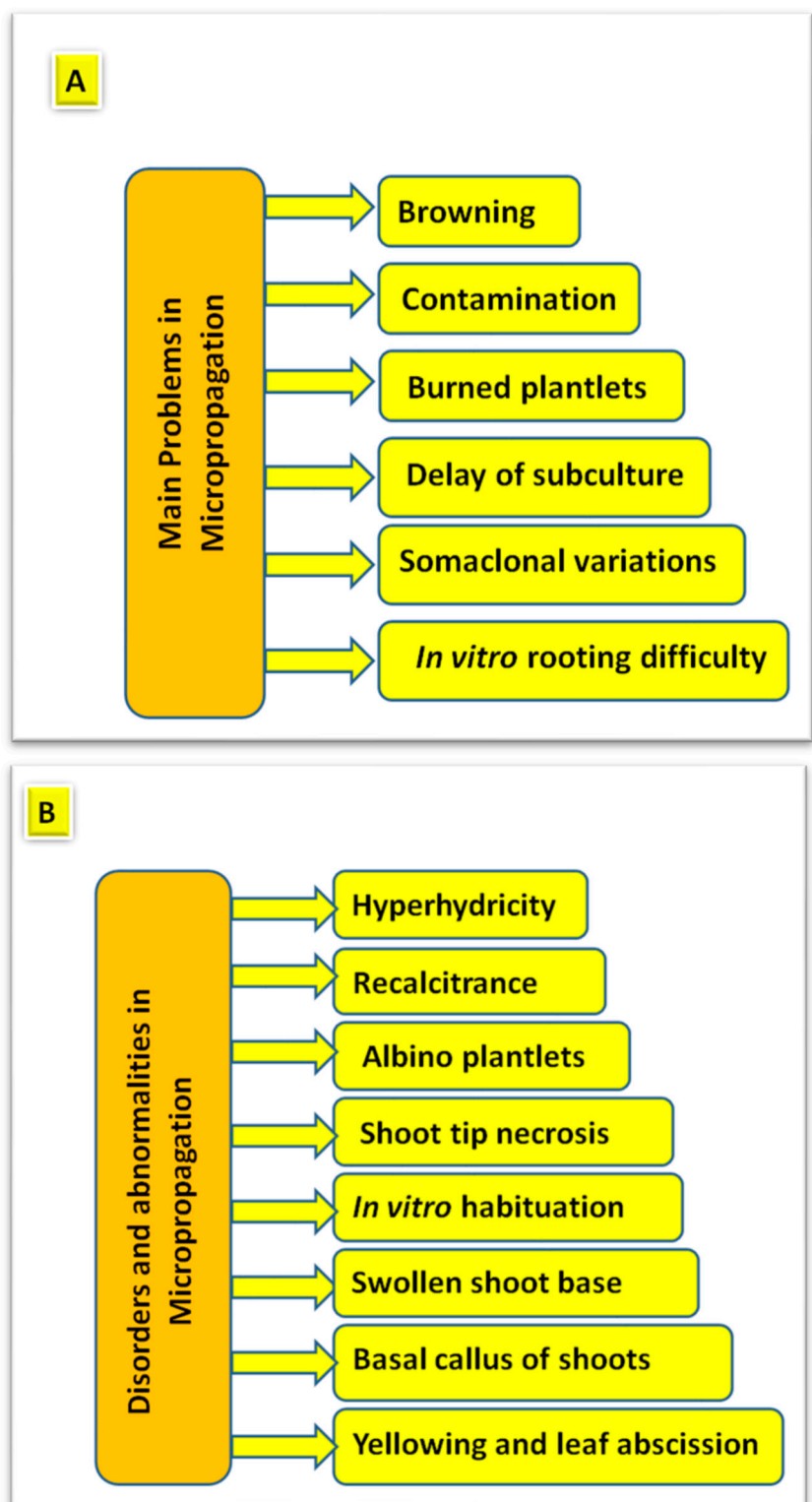

**Figure 11.** A chart showing the main problems in the field of plant micropropagation (**A**) and different disorders and abnormalities (**B**) in this field.

## 6. Conclusions

Micropropagation permits rapid production of a disease-free, uniform, multiplied planting material, with high quality irrespective of the season and weather. However, *in vitro* techniques are very expensive, since skilled manual labor and specific equipment are needed plus there is the high price of plant materials. So, the efficiency of the *in vitro* protocol should

always be optimized and improved when possible. Many problems and abnormalities that hinder the application of *in vitro* micropropagation, mainly in PTC laboratories, have been discussed along with suitable remedies for these obstacles. This is an overview to highlight the severity of the production problems in PTC and to suggest different solutions of alleviating or combating the anomalies. Let us put into perspective not only the past and present knowledge of the various topics but also the future directions to be reviewed. This may contribute toward more understanding of the obstacles production of tissue-cultured plants and their suitable solutions. As far as we know, this is the first report that includes the challenges of plant micropropagation in PTC laboratories and has suggested solutions for those. Further investigations are needed in the future for those laboratories that suffer from a lack of facilities and technical expertise, which could help in an increase in plant productivity using tissue-culture techniques to meet the growing food needs of the population, in light of the scarcity of arable land and water, especially due to climate change. Therefore, the technical and financial management of PTC laboratories is needed to overcome the previous challenges facing the field of plant-tissue culture.

**Author Contributions:** This main idea of this work was attributed to M.E.E.-M. Photos were taken by both M.E.E.-M. and N.A. The first draft was achieved by H.E.-R. and N.A. The manuscript was edited and revised by J.D., M.K.S., N.T., Y.B., T.A.S. All authors contributed in writing the manuscript and interpreting the information presented. All authors have read and agreed to the published version of the manuscript.

**Funding:** Project no. TKP2021-EGA-20 (Biotechnology) has been implemented with the support provided from the National Research, Development and Innovation Fund of Hungary, financed under the TKP2021-EGA funding scheme. Neama Abdalla thanks the Hungarian Tempus Public Foundation (TPF), Bilateral State Scholarships, grant no. AK-00184-003/2021 for financializing and supporting this work.

**Acknowledgments:** The authors greatly thank the Plant Biotechnology Department, Biotechnology Research Institute, National Research Centre. The authors also thank the staff members of the Physiology and Breeding of Horticultural Crops Laboratory, Department of Horticulture, Faculty of Agriculture, Kafrelsheikh University, Kafr El-Sheikh, Egypt for supporting the completing this work.

**Conflicts of Interest:** The authors declare that there are no conflicts of interest.

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
