# Peer review of "An Academic and Technical Overview on Plant Micropropagation Challenges"

_horticulturae, doi:10.3390/horticulturae8080677_

Round 1
Reviewer 1 Report
The manuscript “Plant Tissue Culture Challenges, Problems, Disorders and Abnormalities: A Special Concern on Developing Countries Laboratories”, from Neama Abdalla and coworkers, is a review concerning the production of micropropagated plants in plant tissue culture laboratories. The applying of plant tissue culture techniques for commercial propagation of plants faces technical, biological, physiological and genetical obstacles. In developing countries, these obstacles are due to a lack of facilities and professional technicians. This study aims to show problems of plant tissue culture, their scientific explanation and it suggests solutions for developing countries, where facilities are often incomplete or with low technical quality and with scarce funds for sustainable production of food for their population.
Although the topic deserves great attention, however this work is extremely ambitious to be collected in a single review. This is the reason why, in my opinion, the work is extremely superficial and does not touch many topics concerning basic micropropagation, therefore it is unable to significantly contribute to the field, neither scientifically nor technically.
According to me, this work sounds more appropriate for an excellent informative periodical journal rather than a technical/scientific journal.
I would suggest to the authors to better focus one of the topics of their review and to deepen and discuss the chosen argument.
Best whishes
Author Response
Reviewer 1#
Comments and Suggestions for Authors
The manuscript “Plant Tissue Culture Challenges, Problems, Disorders and Abnormalities: A Special Concern on Developing Countries Laboratories”, from Neama Abdalla and coworkers, is a review concerning the production of micropropagated plants in plant tissue culture laboratories. The applying of plant tissue culture techniques for commercial propagation of plants faces technical, biological, physiological and genetical obstacles. In developing countries, these obstacles are due to a lack of facilities and professional technicians. This study aims to show problems of plant tissue culture, their scientific explanation and it suggests solutions for developing countries, where facilities are often incomplete or with low technical quality and with scarce funds for sustainable production of food for their population.
Response: Yes, many thanks for your comment. We focused on the plant micropropagation according to comment from the editor. Many changes in the revised MS were added beside sections on the methodology of the review, to highlight how we establish and built the MS, their sources, how we thought about the sections in this MS and the TOC as well! Another section on the General discussion for justification why this MS was edited with more explanations on this MS by asking some questions and answering these questions to present the importance of this work!
Although the topic deserves great attention, however this work is extremely ambitious to be collected in a single review. This is the reason why, in my opinion, the work is extremely superficial and does not touch many topics concerning basic micropropagation, therefore it is unable to significantly contribute to the field, neither scientifically nor technically.
According to me, this work sounds more appropriate for an excellent informative periodical journal rather than a technical/scientific journal.
Response: We totally respect your comment, but with all due respect we disagree with your opinion for these following reasons:
(1)- this MS according to opinion of the other 2 reviewers is good as they said:
Reviewer 2# “The authors have made a good effort to review the most important factors influencing the micropropagation of horticultural plants”
Reviewer 3# “The review is really interesting. The paper gives a good overview of the current state of knowledge about the problems encountered during micropropagation of many plant species, also indicating possibilities to avoid or solve these problems.”
(2)- This review depended on different sources of published materials in high impacted journals or sound other published materials beside some sources from the experience for more than 25 years of some authors in their academic and practical field from their working on universities and commercial companies in the field of plant biotechnology!
(3)- this work after revision contains more deep discussion and more explanations even in tables or drawn figures or within the text in the revised MS.
I would suggest to the authors to better focus one of the topics of their review and to deepen and discuss the chosen argument.
Best whishes
Response: We totally respect this comment, but with all due respect we disagree with your comment for these reasons:
(1)- if we discuss only one topic in the revised MS, that means the MS will be like many published MSs and then lost the importance and strength of this MS as a first report including all these topics in only one MS!!!
(2)- The revised MS now is better compared to the first round as we have already changed in different parts in the revised MS and added 2 important sections, deep discussion in parts, more details in some tables and new figures were drawn!
So, please accept our reasoning and the new, revised version of the manuscript, many thanks in advance!

Reviewer 2 Report
The authors have made a good effort to review the most important factors influencing the micropropagation of horticultural plants. In my opinion, they could provide some more specific details for solving some difficulties during plant micropropagation. In addition, the authors did not write the references section according to the instructions. Please check all the references.
For example:
L595-597. The reference is from a chapter in a book.
Author 1, A.; Author 2, B. Title of the chapter. In Book Title, 2nd ed.; Editor 1, A., Editor 2, B., Eds.; Publisher: Publisher Location, Country, Year; Volume 3, pp. 154–196.
L598-599. The reference is from a book.
Author 1, A.; Author 2, B. Book Title, 3rd ed.; Publisher: Publisher Location, Country, Year; pp. 154–196.
(e.g. Smith, A.B. Textbook of Organic Chemistry; D. C. Jones: New York, NY, USA, 1961; pp 123-126.)
L600-602. The reference is from a journal article.
Author 1, A.B.; Author 2, C.D. Title of the article. Abbreviated Journal Name Year, Volume, page range.
(e.g. Bowman, C.M.; Landee, F.A.; Reslock, M.A. Chemically Oriented Storage and Retrieval System. 1. Storage and Verification of Structural Information. J. Chem. Doc. 1967, 7, 43-47; DOI:10.1021/c160024a013.).
Author Response
Reviewer 2#
Comments and Suggestions for Authors
The authors have made a good effort to review the most important factors influencing the micropropagation of horticultural plants.
Response: Many thanks for your great comment and encouraging words!
In my opinion, they could provide some more specific details for solving some difficulties during plant micropropagation.
Response: Many thanks for your great comment, we have already added many parts in nearly all sections of the revised MS and more details in tables, drawn figures!
Many changes in the revised MS were added beside sections on the methodology of the review, to highlight how we establish and built the MS, their sources, how we thought about the sections in this MS and the TOC as well! and other section on the General discussion for justification why this MS was edited with more explanations on this MS by asking some questions and answering these questions to present the importance of this work!
In addition, the authors did not write the references section according to the instructions. Please check all the references. For example:
L595-597. The reference is from a chapter in a book.
Author 1, A.; Author 2, B. Title of the chapter. In Book Title, 2nd ed.; Editor 1, A., Editor 2, B., Eds.; Publisher: Publisher Location, Country, Year; Volume 3, pp. 154–196.
L598-599. The reference is from a book.
Author 1, A.; Author 2, B. Book Title, 3rd ed.; Publisher: Publisher Location, Country, Year; pp. 154–196.
(e.g. Smith, A.B. Textbook of Organic Chemistry; D. C. Jones: New York, NY, USA, 1961; pp 123-126.)
L600-602. The reference is from a journal article.
Author 1, A.B.; Author 2, C.D. Title of the article. Abbreviated Journal Name Year, Volume, page range.
(e.g. Bowman, C.M.; Landee, F.A.; Reslock, M.A. Chemically Oriented Storage and Retrieval System. 1. Storage and Verification of Structural Information. J. Chem. Doc. 1967, 7, 43-47; DOI:10.1021/c160024a013.).
Response: Many thanks for your great comment!
All list of refs. was changed into the instructions of the journal, thank you again!

Reviewer 3 Report
The review is really interesting. The paper gives a good overview of the current state of knowledge about the problems encountered during micropropagation of many plant species, also indicating possibilities to avoid or solve these problems. At the same time, I believe that the paper should emphasise and possibly supplement the information regarding the possibilities of overcoming problems in laboratories in developing countries.
The names of the plant species on which the cited research was carried out should be given in the text wherever possible, including the Latin names and relevant botanical denominations.
All plant growth regulators, substances and the names of nutrient solutions as well as procedures should be named by their full name with the abbreviation in brackets when first used in the text, so that only abbreviations can be used afterwards. Please check carefully all the text.
Please check also list of the literature cited. In many cases some dots, coma or spaces are missing. Also the Latin names of the plants are non italised in some cases.
Other comments and corrections are directly on the pdf file.

Author Response
Reviewer 3#
Comments and Suggestions for Authors
The review is really interesting. The paper gives a good overview of the current state of knowledge about the problems encountered during micropropagation of many plant species, also indicating possibilities to avoid or solve these problems.
Response: Many thanks for your great comment and encouraging words!!!
At the same time, I believe that the paper should emphasize and possibly supplement the information regarding the possibilities of overcoming problems in laboratories in developing countries.
Response: Many thanks for your great comment again, we added and changes the MS according your important comments, you can find this change in the revised MS!
The names of the plant species on which the cited research was carried out should be given in the text wherever possible, including the Latin names and relevant botanical denominations.
Response: Many thanks for your comment and all corrections for any plant in the revised MS was well-done!!!
All plant growth regulators, substances and the names of nutrient solutions as well as procedures should be named by their full name with the abbreviation in brackets when first used in the text, so that only abbreviations can be used afterwards. Please check carefully all the text.
Response: Many thanks for your comment and all corrections were done, thanks again!!!
Please check also list of the literature cited. In many cases some dots, coma or spaces are missing. Also the Latin names of the plants are non italised in some cases.
Response: Many thanks for your comment and all corrections were done, thanks again!!!
Other comments and corrections are directly on the pdf file.
Response: Many thanks for your comment and all corrections were done in the following table one by one, thanks again!!!

Round 2
Reviewer 1 Report
I have carefully read the revised manuscript and the authors' responses to my comments. The previous version of the manuscript was good from the beginning, and it has been even improved in the present version. However, I respectfully maintain my personal opinion, deriving from my 15 years experience in the field of in vitro cultures of higher tree plants in a specialized research team in the field since the 70s, that the work is aimed at a beginners audience that knows little about this subject and wishes to inquire about micropropagation, rather than to an audience of expert academic or commercial micropropagators who are struggling with and must overcome a micropropagative problem.
This is not a bad thing in itself; it is an Editors’ choice deciding whether to consider educational works worthy of publication.
My best whishes